# MVP-N: A Dataset and Benchmark for Real-World Multi-View Object Classification

**Ren Wang**
Seoul National University
`wangren@capp.snu.ac.kr`

**Jiayue Wang**
Seoul National University
`wangjiayue@capp.snu.ac.kr`

**Tae Sung Kim**
Sun Moon University
`ts7kim@sunmoon.ac.kr`

**Jin-Sung Kim**[*]
Sun Moon University
`jinsungk@sunmoon.ac.kr`

**Hyuk-Jae Lee**[*]
Seoul National University
`hjlee@capp.snu.ac.kr`

## Abstract

Combining information from multiple views is essential for discriminating similar objects. However, existing datasets for multi-view object classification have several limitations, such as synthetic and coarse-grained objects, no validation split for hyperparameter tuning, and a lack of view-level information quantity annotations for analyzing multi-view-based methods. To address this issue, this study proposes a new dataset, MVP-N[2], which contains 44 retail products, 16k real captured views with human-perceived information quantity annotations, and 9k multi-view sets. The fine-grained categorization of objects naturally generates multi-view label noise owing to the inter-class view similarity, allowing the study of learning from noisy labels in the multi-view case. Moreover, this study benchmarks four multi-view-based feature aggregation methods and twelve soft label methods on MVP-N. Experimental results show that MVP-N will be a valuable resource for facilitating the development of real-world multi-view object classification methods. The dataset and code are publicly available at `https://github.com/SMNUResearch/MVP-N`.

## 1 Introduction

Humans live in a three-dimensional (3D) environment comprising various 3D objects with rich information, including shape, color, texture, and size. Human visual perception of 3D objects relies on two-dimensional (2D) observations from different perspectives. Since single-view representations may not provide discriminative features between similar objects, multi-view representations, which combine information from multiple views, are preferred.

Recent multi-view-based methods [1, 6, 28, 30, 35, 41] aggregate multi-view features extracted from well-established single-view classifiers [45, 46, 47, 48, 49, 50] and achieve superior performance. However, some of the settings in the existing methods may not be practical for real-world multi-view object classification. In real-world scenarios, objects are typically observed from diverse viewpoints, and the spatial relationships between cameras and objects are not as easily acquired as in virtual capture environments. Therefore, practical multi-view-based methods should satisfy three properties: P1) Arbitrary numbers of input views are allowed in both the training and test stages. P2) The spatial relationships between the cameras and objects, such as camera positions and relative poses, are not utilized. P3) Views can be obtained from arbitrary viewpoints and permuted randomly rather than

---

[*]Corresponding Authors.

[2]**M**ulti-**V**iew, Retail **P**roducts, Label **N**oise

36th Conference on Neural Information Processing Systems (NeurIPS 2022) Track on Datasets and Benchmarks.

| Method | Year | Main Components | P1 | P2 | P3 | Implementation |
|---|---|---|---|---|---|---|
| *Two-stage:* | | | | | | |
| MVCNN [1] | 2015 | element-wise maximum view-pooling | ✓ | ✓ | ✓ | MATLAB |
| Pairwise Network [2] | 2016 | image sequence decomposition, pairwise learning, weighted view pairs | ✓ | ✗ | ✓ | N/A |
| GIFT [3, 4] | 2017 | inverted file, multi-view matching | ✓ | ✓ | ✓ | N/A |
| RCPCNN [5] | 2017 | view similarity graph, dominant sets, recurrent clustering & pooling | ✓ | ✓ | ✓ | MATLAB |
| GVCNN [6] | 2018 | raw view descriptor, grouping module, intra-group pooling, group fusion | ✓ | ✓ | ✓ | PyTorch† |
| MHBN [7] | 2018 | harmonized bilinear pooling | ✓ | ✓ | ✓ | PyTorch† |
| VERAM [8] | 2018 | observation subnetwork, LSTM, view estimation, reinforcement learning | ✓ | ✗ | ✓ | Torch |
| SeqViews2SeqLabels [9] | 2018 | encoder-RNN, decoder-RNN, attention mechanism | ✓ | ✓ | ✗ | TensorFlow |
| MVCNN-new [10] | 2018 | element-wise maximum view-pooling | ✓ | ✓ | ✓ | PyTorch |
| MV-LSTM [11] | 2018 | bidirectional LSTM, sequence voting, highway network | ✓ | ✓ | ✗ | N/A |
| DeepCCFV [12] | 2019 | DropMax block | ✓ | ✓ | ✓ | N/A |
| MLVCNN [13] | 2019 | loop normalization, LSTM | ✗ | ✓ | ✗ | N/A |
| 3D2SeqViews [14] | 2019 | view feature encoding, hierarchical attention aggregation | ✓ | ✓ | ✗ | N/A |
| RotationNet [15, 16] | 2019 | latent viewpoint variables, pose alignment, view-specific category likelihood | ✗ | ✓ | ✗ | Caffe |
| 3DViewGraph [17] | 2019 | latent semantic mapping, spatial pattern correlation, attentioned correlation aggregation | ✗ | ✗ | ✗ | N/A |
| MVSG-DNN [18] | 2019 | multi-view saliency modeling, LSTM | ✓ | ✓ | ✓ | N/A |
| EMVN [19] | 2019 | group convolution, log-polar transform, homogeneous space convolution, filter localization | ✗ | ✗ | ✗ | PyTorch |
| Relation Network [20] | 2019 | reinforcing block, self-attention mechanism | ✓ | ✓ | ✓ | N/A |
| View-GCN [21] | 2020 | view-graph, local graph convolution, non-local message passing, selective view-sampling | ✗ | ✗ | ✓ | PyTorch |
| HEAR [22] | 2020 | hybrid attention, multi-granular view pooling, hyperbolic embedding & neural network | ✗ | ✗ | ✗ | N/A |
| DRCNN [23] | 2020 | multi-view features rearrangement, affine transformation, dynamic routing | ✗ | ✓ | ✓ | N/A |
| VMM [24] | 2020 | view mixture model, neural expectation maximization, latent view alignment | ✗ | ✓ | ✓ | PyTorch |
| JointMVCNN [25] | 2020 | inter-view information calculation, adaptive-weighting loss fusion | ✗ | ✓ | ✓ | N/A |
| MVLADN [26] | 2021 | set-to-set matching kernel, kernel embedding, harmonized bilinear pooling, VLAD | ✓ | ✓ | ✓ | N/A |
| DRMV [27] | 2021 | feature disentanglement, view permutation consistency regularization, gradient reverse layer | ✗ | ✓ | ✓ | N/A |
| DAN [28] | 2021 | deep-attention network, self-attention mechanism | ✓ | ✓ | ✓ | PyTorch |
| CAR-Net [29] | 2021 | view-wise feature representation & refinement, correspondence-aware representation learning | ✓ | ✗ | ✓ | N/A |
| CVR [30] | 2021 | feature encoder, canonical view representation & aggregator & feature separation loss | ✓ | ✓ | ✓ | PyTorch |
| SVHAN [31] | 2021 | hierarchical feature aggregation module, selective fusion module | ✓ | ✓ | ✗ | N/A |
| MVT [32] | 2021 | patch & position embedding, local transformer encoder, global transformer encoder | ✗ | ✓ | ✓ | N/A |
| VFMVAC [33] | 2022 | voting-based view filtering, cross-view channel shuffle, aggregating convolution | ✗ | ✓ | ✗ | N/A |
| *Three-stage (Hypergraph):* | | | | | | |
| iMHL [34] | 2018 | hypergraph construction, inductive multi-hypergraph learning | ✓ | ✓ | ✓ | N/A |
| HGNN [35] | 2019 | hypergraph construction, hypergraph neural network | ✓ | ✓ | ✓ | PyTorch |
| MHGNN [36] | 2021 | hypergraph construction, multi-scale hypergraph neural network | ✓ | ✓ | ✓ | N/A |
| HGNN+ [37] | 2022 | HGNN, hyperedge groups construction & fusion, two-stage hypergraph convolution | ✓ | ✓ | ✓ | PyTorch |
| AMHCN [38] | 2022 | hypergraph construction, adaptive multi-hypergraph convolutional network | ✓ | ✓ | ✓ | N/A |
| GHSC [39] | 2022 | hypergraph with edge-dependent vertex weights, general hypergraph spectral convolution | ✓ | ✓ | ✓ | N/A |
| *Three-stage (Part):* | | | | | | |
| Parts4Feature [40] | 2019 | generally semantic parts, region proposal network, global feature learning | ✓ | ✓ | ✓ | N/A |
| FG3D-Net [41] | 2021 | generally semantic parts, region proposal network, RNN, hierarchical part-view attention | ✓ | ✓ | ✓ | TensorFlow |

Table 1: Summary of 39 multi-view-based feature aggregation methods published from 2015 to August 2022. **N/A**: The source code is not publicly available. '†': Unofficial implementation.

| | RGB-D Object [42] | ModelNet40 [43] | MIRO [15] | ScanObjectNN [44] | FG3D [41] | **MVP-N (ours)** |
|---|---|---|---|---|---|---|
| Year | 2011 | 2015 | 2018 | 2019 | 2021 | 2022 |
| Representation | RGB-D | Mesh | RGB | Point Cloud | Mesh | RGB |
| #Categories | 51 | 40 | 12 | 15 | 66 | 44 |
| Real-world objects | ✓ | ✗ | ✓ | ✓ | ✗ | ✓ |
| Real capture environment | ✓ | ✗ | ✓ | ✗ | ✗ | ✓ |
| Fine-grained | ✗ | ✗ | ✗ | ✗ | ✓ | ✓ |
| Validation set | ✗ | ✗ | ✗ | ✗ | ✗ | ✓ |
| View-level annotation | ✗ | ✗ | ✗ | ✗ | ✗ | ✓ |

Table 2: Comparison of MVP-N and existing multi-view object classification datasets.

predefined view configurations. Table 1 summarizes 39 recent multi-view-based feature aggregation methods with these three properties.

Moreover, as shown in Table 2, existing datasets [42, 43, 15, 44, 41] may not be sufficient for developing practical multi-view-based methods owing to three main limitations: 1) Synthetic polygon mesh objects and coarse-grained categorization. 2) No validation split causes hyperparameter tuning directly on the test set. 3) The lack of view-level information quantity annotations makes it difficult to interpret how informative and uninformative views are utilized for discriminating objects.

To resolve the above limitations, this study proposes MVP-N, a new dataset containing 9k multi-view sets constructed from 16k real captured views of 44 real-world fine-grained retail products. In MVP-N, different objects can appear similar or identical in specific views, referred to as high inter-class view similarity. In this case, humans cannot classify an object accurately from these views for fine-grained (instance-level) object categorization, making the classification task challenging. Here, views with human uncertainty of class labels are denoted as uninformative views. The inconsistency between the one-hot manner of class labels and human judgment causes multi-view label noise. Soft label methods [51, 52, 53, 54, 55, 56, 57, 58, 59, 60] can help alleviate the inconsistency and render view-level predictions more consistent with human judgments, allowing the study of learning from noisy labels [61, 62] in the multi-view case.

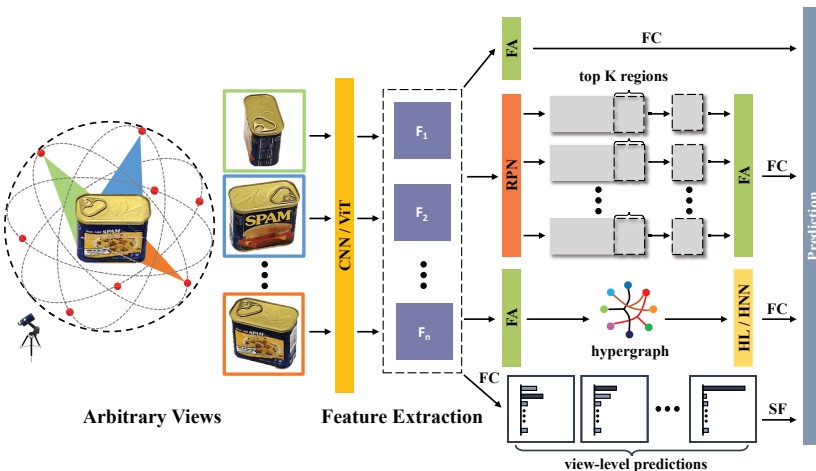

Figure 1: Illustration of the framework for multi-view object classification. **FA**: Feature Aggregation. **RPN**: Region Proposal Network. **FC**: Fully Connected Layer. **HL**: Hypergraph Learning. **HNN**: Hypergraph Neural Network. **SF**: Score Fusion.

MVP-N provides a human-perceived information quantity (HPIQ) annotation for each view in the train/valid/test split, which is defined as informative or uninformative. HPIQ annotations can help analyze how multi-view-based feature aggregation methods discriminate similar objects. Furthermore, this study proposes a new metric and an evaluation protocol based on HPIQ annotations to evaluate the performance of soft label methods for multi-view object classification.

The contributions of this study can be summarized as follows:

- A real-world multi-view fine-grained dataset with HPIQ annotations is proposed to resolve the limitations of existing datasets for developing practical multi-view-based methods.

- Recent multi-view-based feature aggregation methods are comprehensively summarized regarding their practicability in real-world scenarios.

- A new metric and an evaluation protocol are proposed based on HPIQ annotations to evaluate the performance of soft label methods in the multi-view case.

- Four multi-view-based feature aggregation methods and twelve soft label methods are benchmarked on MVP-N, introducing new findings.

The rest of this paper is organized as follows. Section 2 briefly reviews recent multi-view-based feature aggregation methods, existing datasets, and soft label methods. Sections 3 and 4 present the details of MVP-N and the benchmark, respectively. Section 5 presents the analysis results. Section 6 concludes the study and discusses its broader impact.

## 2 Related Work

**Multi-view-based feature aggregation.** As illustrated in Figure 1, existing methods can be categorized into two-stage and three-stage strategies. Two-stage strategies render 2D images from different viewpoints of a 3D object and then perform classification by aggregating multi-view features extracted from 2D convolutional neural networks (CNNs) [45, 46, 47, 48, 49] or vision transformers [50]. The pioneering work, MVCNN [1], aggregates multi-view features in a view-pooling layer based on the element-wise maximum operation. MVCNN is straightforward but causes information loss in the views, and it treats all views equally without exploiting the relationship among them. Later studies attempt to address this issue and perform better multi-view feature aggregation, such as pooling across similar views [5, 6, 7, 22, 26], weighting information from views via the self-attention mechanism [9, 14, 20, 28, 30, 31, 32], and encoding the spatial relationship among views with known capturing settings [9, 11, 14, 17, 21, 29]. Three-stage strategies can be categorized into

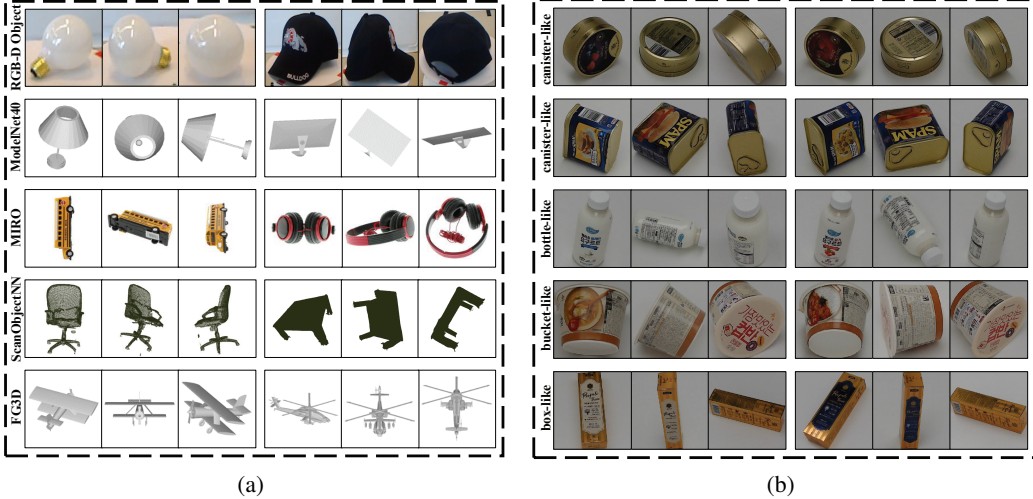

Figure 2: (a) Sample objects in five existing datasets. (b) Sample objects in the proposed MVP-N.

hypergraph-based [34, 35, 36, 37, 38, 39] and part-based [40, 41] methods. Hypergraph-based methods exploit the high-order correlation among objects by constructing a hypergraph on object-level features [1, 6] and help learn discriminative representations from a global perspective. However, most methods [35, 36, 37, 38, 39] require a certain number of test samples to construct the hypergraph, meaning that a single test sample is not allowed in the inference phase. Part-based methods first detect the generally semantic parts (GSPs) from multi-view features via a region proposal network [63] and then perform aggregation on the top K region proposals selected according to their semantic scores. In the training stage of GSPs detection, ground truth GSPs are generated from the 3D shape segmentation benchmarks [64, 65, 66]. Three-stage strategies are more sophisticated than two-stage strategies but may not be practical in real-world scenarios because of the complex inference phase or additional part-level annotations.

**Multi-view object classification datasets.** ModelNet40 [43] is the most commonly used dataset for developing multi-view-based methods. Since its objects are synthetic and coarse-grained, some research [15, 16, 21, 41, 30, 33] selectively uses other datasets for further evaluation, including RGB-D Object [42], MIRO [15], ScanObjectNN [44], and FG3D [41]. The performance of multi-view-based methods on the above datasets is provided in Appendix A. Figure 2 shows a comparison of objects between the existing datasets and MVP-N.

**Soft label.** Soft label methods can be categorized into implicit regularization, human annotation, and label refurbishment. Implicit regularization [54] softens labels by taking an average with a uniform distribution over one-hot labels. Human annotation [60] uses the distribution of human categorization judgments to construct soft labels. Label refurbishment [51, 52, 53, 56, 55, 57, 58, 59] generates soft labels via a weighted combination of one-hot labels and model predictions. The main differences among label refurbishment methods are how to utilize model predictions, class-level or image-level soft labels, the period of supervision with soft labels, and the weighting factor. Unlike general single-image classification tasks [67, 68, 69, 70, 71, 72], this study validates the robustness of soft label methods for multi-view object classification.

## 3 MVP-N: Dataset Design and Construction

**Object selection.** Retail products can be distinguished without semantic confusion[3]. Therefore, a fine-grained categorization [73] can be easily established. Furthermore, retail products of the same brand with different flavors provide high inter-class view similarity. In total, 44 retail products from

---

[3]Ma et al. [11] and Chen et al. [8] point out the semantic confusion issue in ModelNet40 [43].

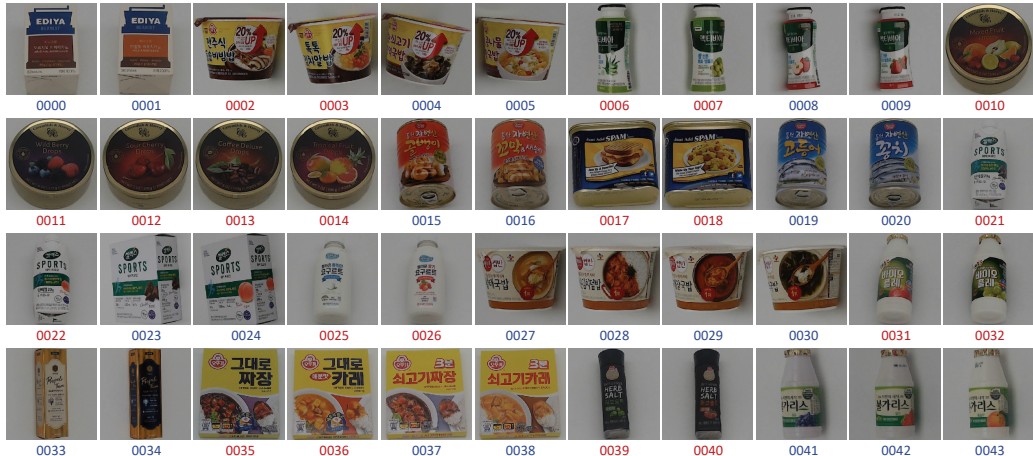

Figure 3: List of 44 objects in MVP-N. Nineteen groups of similar-looking objects are illustrated.

16 brands are selected. Each brand contains two to five objects with similar appearances. Figure 3 lists the selected objects.

**Data collection.**   A multi-camera setup is built to collect multi-view images. This design enables a sizable and closed capture space with a clean white background. Twenty cameras (Logitech StreamCam[4]) are mounted above the tabletop and around the workspace. All cameras are well-calibrated and have manual focus. Lighting equipment is installed above the setup to illuminate the object and its surroundings. Figure 4(a) shows the real capture environment and camera configurations. To provide different distributions of views in the train/valid/test split, we collect multi-view images in two ways, as follows:

- *Collection A* (24 trials in the 12-camera configuration): The object is placed at the center of the tabletop using a predefined pose and rotation. Here, six poses (top, bottom, front, back, left, and right) and four rotations ($0°$, $90°$, $180°$, and $270°$) are defined. With 24 trials and 12 cameras, 288 images of the object are captured.

- *Collection B* (50 trials in the 16-camera configuration): The object is randomly placed on the tabletop, and its pose and rotation are set randomly for each trial. With 50 trials and 16 cameras, 800 images of the object are captured.

**Data annotation.**   We hire ten well-trained annotators with more than six months of experience in image classification and bounding box annotations. The annotators first observe[5] the objects and then group them based on their similarity in appearance. Consequently, 19 groups are assigned. Next, the captured images are distributed equally among the annotators. Each image is annotated once by a single annotator. Since the class label is automatically obtained when capturing an image of a single object, annotators draw a bounding box to enclose the foreground object and judge the quantity of information in its view. The following three options are provided for information quantity judgment:

- *Sufficiently informative*: A distinctive appearance is sufficiently included in this view. The object can be classified correctly without additional information from other views.

- *Less informative*: A distinctive appearance is partially included in this view. High classification accuracy cannot be guaranteed using only this view.

- *Uninformative*: A distinctive appearance is not included in this view. Additional information from other views is required to classify the object correctly.

Figure 4(b) shows examples of human judgments on information quantity. The average annotation time per image is 45 s.

---

[4] https://www.logitech.com/en-us/products/webcams/streamcam.960-001289.html

[5] Text recognition of the product packaging is not considered here.

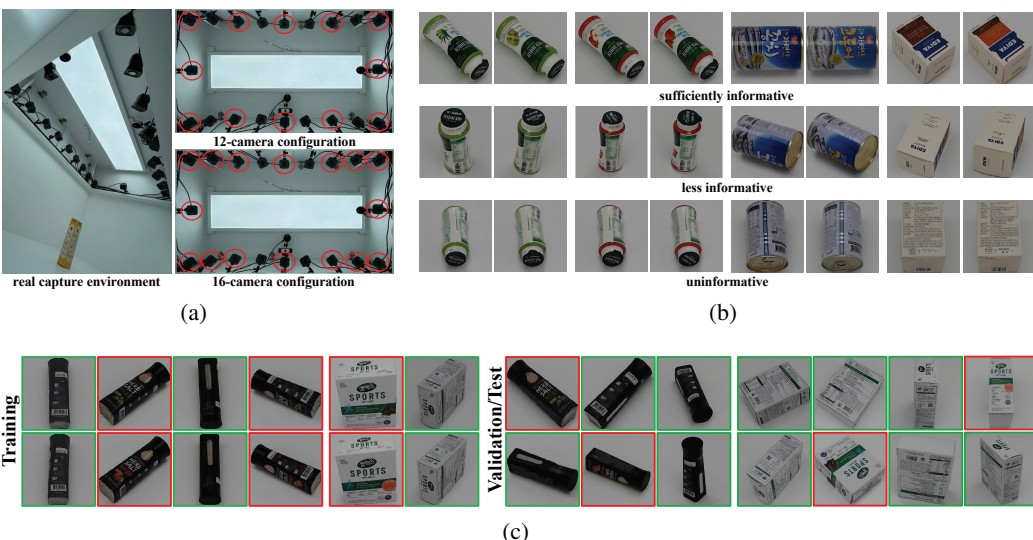

Figure 4: Illustration of the construction for MVP-N. (a) The real capture environment and camera configurations. (b) Examples of human judgments on information quantity. (c) Examples of multi-view sets in the train and validation/test sets. The informative and uninformative views are respectively marked with red and green rectangles.

**Quality control and data filtering.** Two researchers check the quality of the bounding box annotations. Subsequently, the annotators correct the missed and imprecise ones. To guarantee the quality of 'informative/uninformative' (HPIQ) annotations, images with 'less informative' annotations are first filtered. Since a single annotation per image may introduce an annotation bias, two researchers and a volunteer[6] provide three additional information quantity annotations for the remaining images. Subsequently, images with annotation disagreements are filtered. Each remaining image has one HPIQ and one bounding box annotation.

**Data preprocessing.** The resolution of captured images is $1920 \times 1080$. Since the commonly used resolution of views in existing datasets is $224 \times 224$, each image is cropped based on the center point of its bounding box annotation. Specifically, if the width and height of the bounding box are less than 224 pixels, the image is directly cropped to $224 \times 224$. Otherwise, it is cropped to the maximum value between the width and height of the bounding box and then resized to $224 \times 224$. This step removes most of the background and leaves the entire foreground object with its original aspect ratio.

**Train/valid/test split and design rationalization.** The design scheme comprises view sampling and multi-view set construction. Multi-view sets are constructed by combining the sampled views. For the training set, 20 informative and 20 uninformative views are manually sampled from *Collection A* for each object to cover its overall appearance. Subsequently, ten multi-view sets are constructed for each object. Each set has two to six views, containing at least one informative and one uninformative view. For the validation and test sets, 40 informative and 120 uninformative views are randomly sampled from *Collections A and B* for each object. Subsequently, 100 multi-view sets are constructed for each object. Each set has two to six views, containing only a single informative view. No multi-view set shares more than two views with others to guarantee the diversity of view combinations. Examples of the multi-view sets are shown in Figure 4(c). The training set design enables sufficient learning. A recent study [30] uses a similar design[7] and achieves approximately 90% category-level accuracy on RGB-D Object [42]. Compared with the training set, the large-scale validation and test sets provide diverse view combinations with different proportions of informative views, thus reducing the test bias caused by limited multi-view sets. Since all multi-view sets contain at least one informative view, the human accuracy on MVP-N is 100%.

---

[6]Graduate student majoring in Korean Language and Literature.

[7]The training set comprises 40 sampled views and five multi-view sets on average per category.

# 4 Benchmark on MVP-N

**Method selection.** The selection criteria of the benchmark methods are representativeness, practicality in real-world scenarios, and availability of the source code. Multi-View CNN (MVCNN-new) [10], Group-View CNN (GVCNN) [6], Deep-Attention Network (DAN) [28], and Canonical View Representation (CVR) [30] are selected from the multi-view-based feature aggregation methods. Knowledge Distillation (KD) [51], Soft Bootstrapping (SB) [52], Hard Bootstrapping (HB) [52], Label Smoothing (LS) [54], Dynamic Soft Bootstrapping (DSB) [53], Dynamic Hard Bootstrapping (DHB) [53], Self-Adaptive Training (SAT) [56], Likelihood Ratio Test (LRT) [55], Self-Evolution Average Label (SEAL) [57], Progressive Label Correction (PLC) [58], Online Label Smoothing (OLS) [59] are selected from the soft label methods. HPIQ uses hard labels for informative views and uniform distributions over the classes from the same group as soft labels for uninformative views. The baselines are MVCNN-new and standard cross entropy with hard labels (CE). Additionally, a hyperparameter search is performed for each method according to the relevant sensitivity analysis in its original paper. Appendix B provides the results of the hyperparameter search and the best configuration for each method.

**Experimental Platform.** The experiments are performed on four NVIDIA GeForce RTX 3090 GPUs and an Intel(R) Core(TM) i9-10900X CPU @ 3.70GHz. The models are implemented in PyTorch [74] and trained with CUDA 11.1 and cuDNN 8 as the computational back-ends.

**Training details.** ResNet-18 [49] pre-trained on ImageNet [68] is adopted as the backbone for computational efficiency. The network is trained using the stochastic gradient descent (SGD) optimizer with a momentum of 0.9 and a weight decay of $10^{-3}$. The training scheme of the two-stage feature aggregation methods comprises single- and multi-view training. In single-view training, the network is trained as general single-image classification tasks for 30 epochs with a batch size of 128. The initial learning rate is $10^{-2}$ and reduced by half every 10 epochs. In multi-view training, the variable-length views are first zero-padded to the maximum number of views and then batched together. The network is trained with the feature aggregation module for 50 epochs with one warm-up [75] epoch and a batch size of 32. The initial learning rate is $10^{-3}$ and gradually decreases to zero through the cosine annealing strategy [76]. The training details of the soft label methods are identical to those of single-view training. All methods are run five times with various random seeds.

**Evaluation of multi-view-based feature aggregation methods.** Multi-view accuracy (MVA), mean confidence for correct predictions (MCC), and mean confidence for wrong predictions (MCW) are used for performance evaluation and reported as the mean and standard deviation over five trials. The model size, number of floating-point operations (FLOPs), and inference latency are measured to evaluate the computational efficiency. The model size is obtained by calculating the number of trainable parameters. The number of input views is set to six to measure the FLOPs and inference latency. The inference latency is measured on a single GPU with CUDA and cuDNN and reported as the mean and standard deviation over 500 trials.

**Evaluation of soft label methods.** The proposed evaluation protocol is based on HPIQ annotations. For informative views, single-view accuracy (SVAI), mean confidence for correct predictions (MCCI), and mean confidence for wrong predictions (MCWI) are used for performance evaluation. As for uninformative views, even humans cannot always make correct predictions because of multi-view label noise. Therefore, the single-view accuracy and mean confidence for correct/wrong predictions are inappropriate for performance evaluation. In this study, mean confidence difference between predictions and ground truths (MCDU) is proposed as a new metric for performance evaluation on uninformative views. The following formula gives MCDU:

$$MCDU = \frac{1}{N}\sum_{i=1}^{N}(\max(p_i) - p_i(y_i)) \tag{1}$$

where $N$ is the number of wrong predictions for all uninformative views in the validation or test set, $p_i$ is the predicted probability distribution, and $y_i$ is the ground-truth class label. When MCDU is high, wrong predictions from uninformative views negatively affect performance. Since predictions from uninformative views are not reliable, a lower MCDU indicates better performance. For the overall performance evaluation, MVA is obtained by combining the predictions via the mean rule [77]. All results are reported as the mean and standard deviation over five trials.

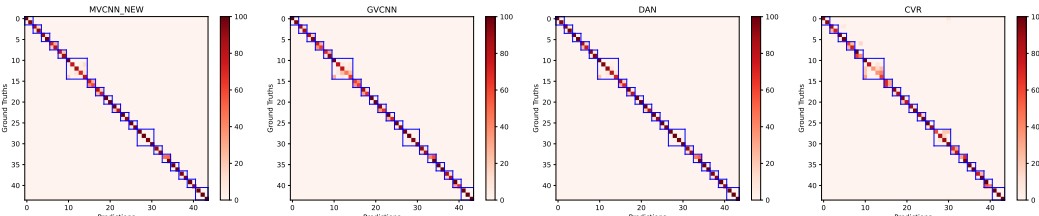

Figure 5: Confusion matrix for four multi-view-based feature aggregation methods. Categories from the same group are marked with blue squares.

| Method | SVA (%) | SVAI (%) ↑ | MCCI ↑ | MCWI ↓ | MCDU ↓ | MVA (%) ↑ |
|---|---|---|---|---|---|---|
| *Validation:* | | | | | | |
| CE | 76.76 ± 0.24 | 99.44 ± 0.17 | 0.9475 ± 0.0031 | 0.6076 ± 0.0368 | 0.3977 ± 0.0091 | 83.05 ± 0.56 |
| KD [51] | 78.47 ± 0.55 | 99.62 ± 0.08 | 0.9587 ± 0.0009 | 0.5799 ± 0.0295 | 0.3867 ± 0.0040 | 85.72 ± 1.24 |
| SB [52] | 74.41 ± 0.36 | 99.08 ± 0.33 | 0.8911 ± 0.0074 | 0.5573 ± 0.0177 | 0.2945 ± 0.0046 | 83.31 ± 0.41 |
| HB [52] | 76.69 ± 0.18 | 99.44 ± 0.16 | 0.9469 ± 0.0029 | 0.6073 ± 0.0369 | 0.3989 ± 0.0097 | 82.73 ± 0.60 |
| LS [54] | 76.03 ± 0.36 | 99.26 ± 0.15 | 0.7711 ± 0.0056 | **0.4101 ± 0.0262** | 0.2534 ± 0.0093 | 84.30 ± 1.05 |
| DSB [53] | 76.06 ± 0.98 | 99.15 ± 0.58 | 0.9148 ± 0.0522 | 0.5704 ± 0.0313 | 0.3577 ± 0.0626 | 82.71 ± 0.69 |
| DHB [53] | 76.67 ± 0.27 | 99.48 ± 0.18 | 0.9454 ± 0.0022 | 0.6113 ± 0.0301 | 0.3971 ± 0.0069 | 82.60 ± 0.70 |
| SAT [56] | 74.55 ± 0.40 | 99.18 ± 0.19 | 0.8746 ± 0.0049 | 0.5465 ± 0.0179 | 0.2256 ± 0.0058 | 86.52 ± 0.36 |
| LRT [55] | 76.57 ± 0.52 | 99.60 ± 0.15 | **0.9609 ± 0.0018** | 0.6094 ± 0.0642 | 0.4240 ± 0.0104 | 84.29 ± 1.26 |
| SEAL [57] | 71.97 ± 0.33 | 98.92 ± 0.23 | 0.6846 ± 0.0036 | 0.4379 ± 0.0102 | **0.1404 ± 0.0018** | 85.48 ± 0.65 |
| PLC [58] | 76.51 ± 0.27 | 99.33 ± 0.20 | 0.9469 ± 0.0033 | 0.6126 ± 0.0424 | 0.4042 ± 0.0119 | 82.37 ± 0.72 |
| OLS [59] | 76.63 ± 0.14 | 99.30 ± 0.17 | 0.9336 ± 0.0041 | 0.5852 ± 0.0273 | 0.3774 ± 0.0101 | 82.90 ± 0.57 |
| HPIQ | 62.77 ± 0.42 | **99.73 ± 0.04** | 0.9246 ± 0.0057 | 0.5538 ± 0.0447 | 0.1530 ± 0.0068 | **93.55 ± 0.79** |
| *Test:* | | | | | | |
| CE | 78.65 ± 0.44 | 99.15 ± 0.11 | 0.9383 ± 0.0028 | 0.6035 ± 0.0442 | 0.3892 ± 0.0070 | 83.37 ± 1.05 |
| KD [51] | 80.38 ± 0.24 | 99.49 ± 0.09 | 0.9509 ± 0.0014 | 0.5574 ± 0.0606 | 0.3737 ± 0.0014 | 86.77 ± 1.24 |
| SB [52] | 76.22 ± 0.27 | 98.73 ± 0.15 | 0.8789 ± 0.0068 | 0.5230 ± 0.0210 | 0.2862 ± 0.0107 | 83.85 ± 0.84 |
| HB [52] | 78.52 ± 0.51 | 99.10 ± 0.15 | 0.9376 ± 0.0022 | 0.6050 ± 0.0305 | 0.3899 ± 0.0068 | 83.20 ± 1.11 |
| LS [54] | 77.65 ± 0.30 | 98.82 ± 0.30 | 0.7522 ± 0.0054 | **0.3843 ± 0.0309** | 0.2474 ± 0.0114 | 83.96 ± 1.50 |
| DSB [53] | 77.73 ± 0.81 | 98.90 ± 0.35 | 0.9037 ± 0.0523 | 0.5784 ± 0.0606 | 0.3457 ± 0.0610 | 83.09 ± 0.83 |
| DHB [53] | 78.34 ± 0.46 | 99.07 ± 0.17 | 0.9365 ± 0.0023 | 0.6040 ± 0.0488 | 0.3871 ± 0.0041 | 83.07 ± 0.87 |
| SAT [56] | 76.28 ± 0.43 | 99.00 ± 0.14 | 0.8620 ± 0.0049 | 0.5293 ± 0.0337 | 0.2145 ± 0.0063 | 87.37 ± 1.15 |
| LRT [55] | 77.85 ± 0.46 | 99.33 ± 0.19 | **0.9542 ± 0.0013** | 0.5881 ± 0.0327 | 0.4076 ± 0.0141 | 83.78 ± 2.05 |
| SEAL [57] | 73.58 ± 0.54 | 98.41 ± 0.25 | 0.6674 ± 0.0033 | 0.4018 ± 0.0085 | **0.1326 ± 0.0011** | 86.42 ± 0.74 |
| PLC [58] | 78.40 ± 0.47 | 99.07 ± 0.11 | 0.9383 ± 0.0031 | 0.6070 ± 0.0370 | 0.3948 ± 0.0106 | 82.96 ± 1.06 |
| OLS [59] | 78.40 ± 0.49 | 99.00 ± 0.12 | 0.9225 ± 0.0029 | 0.5799 ± 0.0239 | 0.3684 ± 0.0080 | 83.35 ± 1.05 |
| HPIQ | 63.31 ± 0.38 | **99.68 ± 0.10** | 0.9186 ± 0.0059 | 0.5934 ± 0.0222 | 0.1481 ± 0.0076 | **94.36 ± 0.56** |

Table 3: Performance of twelve soft-label methods. **SVA**: single-view accuracy for all views. '↑': Higher is better. '↓': Lower is better. The best performance is boldfaced.

## 5 Analysis Results

**Error type.** Figure 5 shows the confusion matrix for four multi-view-based feature aggregation methods. The multi-view errors made by the models mostly derive from the same group of similar-looking objects, indicating that the model predictions match well with the grouping scheme made by humans. Moreover, as shown in Table 3, all soft label methods achieve high SVAI (approximately 99%), consistent with human judgments on informative views. The gap between SVA and SVAI indicates that single-view errors mostly originate from uninformative views. Therefore, utilizing information from informative views is crucial for discriminating similar objects.

**Comparison of soft label methods.** As shown in Table 3, HPIQ achieves the best MVA but the lowest SVA. The results indicate that SVAI, MCCI, and MCDU significantly affect MVA since the low accuracy of uninformative views does not degrade the multi-view performance. SAT [56] achieves the second-best MVA because of its relatively high MCCI and low MCDU. SEAL [57] has the best MCDU but the worst MCCI, causing a decrease in MVA compared to SAT. Moreover, since the ground truths of uninformative views are error-free, the methods containing the label flip component, including HB [52], DHB [53], LRT [55], and PLC [58], could not significantly improve

| Method | MVA (%) ↑ | MCC ↑ | MCW ↓ | Model Size (M) ↓ | FLOPs (G) ↓ | Latency (ms) ↓ |
|---|---|---|---|---|---|---|
| *Validation:* | | | | | | |
| MVCNN-new [10] | $89.29 \pm 0.88$ | $\mathbf{0.8812 \pm 0.0040}$ | $0.6568 \pm 0.0120$ | $\mathbf{11.20}$ | $\mathbf{10.91}$ | $\mathbf{6.23 \pm 0.03}$ |
| GVCNN [6] | $85.69 \pm 1.01$ | $0.8275 \pm 0.0044$ | $\mathbf{0.6095 \pm 0.0136}$ | $24.04$ | $10.99$ | $7.60 \pm 0.07$ |
| DAN [28] | $\mathbf{92.05 \pm 0.56}$ | $0.8592 \pm 0.0044$ | $0.6192 \pm 0.0055$ | $17.50$ | $10.95$ | $8.11 \pm 0.04$ |
| CVR [30] | $79.95 \pm 1.89$ | $0.8347 \pm 0.0118$ | $0.6564 \pm 0.0157$ | $34.38$ | $11.08$ | $12.57 \pm 0.07$ |
| *Test:* | | | | | | |
| MVCNN-new [10] | $89.35 \pm 1.21$ | $\mathbf{0.8792 \pm 0.0053}$ | $0.6552 \pm 0.0069$ | $\mathbf{11.20}$ | $\mathbf{10.91}$ | $\mathbf{6.23 \pm 0.03}$ |
| GVCNN [6] | $85.42 \pm 1.37$ | $0.8267 \pm 0.0032$ | $\mathbf{0.6055 \pm 0.0088}$ | $24.04$ | $10.99$ | $7.60 \pm 0.07$ |
| DAN [28] | $\mathbf{91.61 \pm 0.94}$ | $0.8602 \pm 0.0050$ | $0.6211 \pm 0.0062$ | $17.50$ | $10.95$ | $8.11 \pm 0.04$ |
| CVR [30] | $79.99 \pm 2.52$ | $0.8339 \pm 0.0127$ | $0.6457 \pm 0.0166$ | $34.38$ | $11.08$ | $12.57 \pm 0.07$ |

Table 4: Performance of four multi-view-based feature aggregation methods. M, G, and ms denote million, billion, and milliseconds, respectively. '↑': Higher is better. '↓': Lower is better. The best performance is boldfaced. Backbone (ResNet-18): 11.20 M, 11.91 G, and $6.19 \pm 0.05$ ms.

| Method | Validation | | | | | Test | | | | |
|---|---|---|---|---|---|---|---|---|---|---|
| | 2 views | 3 views | 4 views | 5 views | 6 views | 2 views | 3 views | 4 views | 5 views | 6 views |
| *feature aggregation:* | | | | | | | | | | |
| MVCNN-new [10] | $91.93 \pm 0.77$ | $88.80 \pm 0.54$ | $88.30 \pm 1.17$ | $88.25 \pm 1.04$ | $89.18 \pm 1.52$ | $89.52 \pm 1.38$ | $88.98 \pm 1.31$ | $87.36 \pm 1.80$ | $89.09 \pm 1.76$ | $91.82 \pm 1.01$ |
| GVCNN [6] | $93.23 \pm 1.14$ | $84.73 \pm 0.77$ | $84.48 \pm 1.52$ | $81.93 \pm 1.47$ | $84.09 \pm 1.40$ | $89.70 \pm 1.59$ | $82.43 \pm 1.92$ | $84.05 \pm 1.65$ | $82.70 \pm 1.87$ | $88.20 \pm 1.06$ |
| DAN [28] | $93.80 \pm 0.92$ | $91.59 \pm 1.12$ | $91.07 \pm 0.99$ | $\mathbf{91.50 \pm 1.24}$ | $\mathbf{92.32 \pm 0.63}$ | $91.48 \pm 0.44$ | $91.20 \pm 1.17$ | $89.59 \pm 1.12$ | $91.39 \pm 1.51$ | $\mathbf{94.41 \pm 1.01}$ |
| CVR [30] | $86.16 \pm 0.73$ | $81.16 \pm 1.82$ | $79.25 \pm 2.55$ | $76.39 \pm 2.85$ | $76.80 \pm 2.51$ | $84.20 \pm 2.19$ | $79.95 \pm 2.25$ | $77.80 \pm 3.72$ | $77.52 \pm 2.94$ | $80.45 \pm 3.42$ |
| *soft label:* | | | | | | | | | | |
| CE | $92.98 \pm 0.59$ | $82.14 \pm 0.97$ | $80.25 \pm 0.81$ | $79.23 \pm 1.16$ | $80.64 \pm 0.54$ | $90.82 \pm 1.42$ | $79.95 \pm 1.33$ | $79.34 \pm 1.66$ | $80.61 \pm 1.65$ | $86.11 \pm 0.52$ |
| KD [51] | $95.84 \pm 0.47$ | $84.48 \pm 1.03$ | $83.68 \pm 1.99$ | $81.39 \pm 1.68$ | $83.20 \pm 1.71$ | $94.64 \pm 0.85$ | $84.57 \pm 2.49$ | $82.50 \pm 1.33$ | $84.05 \pm 1.31$ | $88.09 \pm 1.82$ |
| SB [52] | $92.23 \pm 0.73$ | $83.77 \pm 0.32$ | $80.55 \pm 0.64$ | $79.73 \pm 1.04$ | $80.30 \pm 0.86$ | $89.11 \pm 1.26$ | $82.57 \pm 0.94$ | $80.91 \pm 1.26$ | $81.09 \pm 1.87$ | $85.59 \pm 0.60$ |
| HB [52] | $92.82 \pm 0.88$ | $81.73 \pm 0.79$ | $80.11 \pm 1.11$ | $78.59 \pm 1.11$ | $80.39 \pm 0.64$ | $90.80 \pm 1.32$ | $79.57 \pm 1.25$ | $79.50 \pm 2.03$ | $80.27 \pm 1.84$ | $85.86 \pm 0.47$ |
| LS [54] | $90.84 \pm 0.84$ | $84.61 \pm 1.35$ | $82.84 \pm 1.65$ | $81.36 \pm 0.76$ | $81.84 \pm 1.74$ | $88.25 \pm 1.32$ | $82.09 \pm 1.32$ | $80.91 \pm 1.78$ | $81.64 \pm 2.22$ | $86.91 \pm 1.02$ |
| DSB [53] | $92.34 \pm 1.78$ | $82.36 \pm 0.94$ | $79.93 \pm 0.44$ | $78.68 \pm 1.11$ | $80.25 \pm 0.85$ | $90.07 \pm 1.28$ | $80.39 \pm 1.33$ | $79.18 \pm 1.39$ | $79.93 \pm 1.38$ | $85.89 \pm 0.85$ |
| DHB [53] | $92.95 \pm 1.02$ | $82.16 \pm 0.53$ | $79.91 \pm 0.57$ | $77.91 \pm 1.36$ | $80.09 \pm 1.13$ | $90.73 \pm 1.40$ | $79.98 \pm 1.44$ | $78.77 \pm 1.31$ | $80.39 \pm 1.17$ | $85.50 \pm 0.65$ |
| SAT [56] | $94.27 \pm 0.18$ | $87.20 \pm 0.68$ | $84.39 \pm 0.96$ | $83.52 \pm 1.12$ | $83.23 \pm 1.10$ | $91.45 \pm 1.21$ | $87.25 \pm 1.08$ | $84.70 \pm 1.76$ | $84.93 \pm 1.98$ | $88.50 \pm 0.86$ |
| LRT [55] | $94.93 \pm 0.60$ | $83.57 \pm 1.25$ | $81.77 \pm 1.40$ | $80.25 \pm 1.93$ | $80.91 \pm 1.77$ | $93.02 \pm 1.19$ | $81.07 \pm 2.98$ | $79.61 \pm 1.97$ | $80.45 \pm 2.34$ | $84.75 \pm 2.21$ |
| SEAL [57] | $93.02 \pm 1.08$ | $86.64 \pm 0.98$ | $84.11 \pm 0.60$ | $81.61 \pm 0.60$ | $82.02 \pm 0.85$ | $90.80 \pm 0.79$ | $85.95 \pm 0.96$ | $83.75 \pm 0.99$ | $84.34 \pm 0.79$ | $87.27 \pm 0.82$ |
| PLC [58] | $92.75 \pm 0.54$ | $81.45 \pm 1.00$ | $79.64 \pm 0.70$ | $78.39 \pm 1.02$ | $79.64 \pm 1.21$ | $90.61 \pm 1.26$ | $79.57 \pm 1.06$ | $79.09 \pm 1.87$ | $80.07 \pm 1.68$ | $85.48 \pm 0.92$ |
| OLS [59] | $92.43 \pm 0.56$ | $81.91 \pm 0.55$ | $80.39 \pm 1.13$ | $79.18 \pm 0.87$ | $80.57 \pm 0.71$ | $90.00 \pm 1.20$ | $80.09 \pm 1.30$ | $79.66 \pm 1.95$ | $80.66 \pm 1.41$ | $86.32 \pm 0.52$ |
| HPIQ | $\mathbf{98.34 \pm 0.36}$ | $\mathbf{95.91 \pm 0.53}$ | $\mathbf{93.23 \pm 1.15}$ | $90.59 \pm 0.97$ | $89.68 \pm 1.88$ | $\mathbf{97.73 \pm 0.38}$ | $\mathbf{96.36 \pm 0.47}$ | $\mathbf{93.64 \pm 1.34}$ | $\mathbf{92.02 \pm 0.86}$ | $92.07 \pm 0.91$ |

Table 5: Influence of the number of uninformative views. The best performance (MVA) is boldfaced.

MCDU. Comparing SB [52] with HB [52] and DSB [53] with DHB [53], using predicted class probabilities as soft labels can significantly reduce MCDU and achieve a higher MVA.

**Comparison of multi-view-based feature aggregation methods.** Table 4 shows the performance of the four selected methods. DAN [28] achieves the best MVA and outperforms MVCNN-new [10] by 2.26%. The self-attention mechanism [78] in DAN helps increase the weight of informative views in the feature aggregation, thus improving MVA and reducing MCW. The decrease in MCC is caused by introducing more correct predictions with less confidence. GVCNN [6] groups the views considering discrimination scores obtained by a raw view descriptor and performs pooling in each group. Features are aggregated with the corresponding weights obtained by the sum of discrimination scores in each group. Since uninformative views are usually grouped owing to similar discrimination scores, the intra-group pooling component helps reduce the uncertainty, causing a lower MCW. As shown in Table 5, in the case of two views, since the group containing a single uninformative view has a relatively small weight, GVCNN achieves a better MVA than MVCNN-new. However, when the group contains more uninformative views, the corresponding weight increases in the feature aggregation, degrading MVA. The performance of CVR [30] is worse than that of the other three methods owing to its main components. CVR transforms multi-view features into a fixed number of canonical view features via an optimal transport solver [79] and then aggregates the features with a transformer encoder [78]. As analyzed in [28], the multi-head attention module in the transformer encoder introduces redundant information for multi-view feature aggregation. As for computation efficiency, except for MVCNN-new, the computational cost from the main components of the other three models is not negligible. For example, the feature aggregation module of CVR triples the model size and slows down the model by half compared with the backbone.

**Influence of the number of uninformative views.** As shown in Table 5, soft label methods are more sensitive to the number of uninformative views, ranging from one to five, than multi-view-based feature aggregation methods. In the case of two views, four soft label methods outperform DAN [28], but all are inferior to DAN in the case of six views. For soft label methods, more uninformative views

yield more uncertainty and may lead to a more significant decrease in MVA. However, when the proportion of uninformative views is small, soft label methods would be a better choice regarding computational efficiency and performance.

**Comparison with existing datasets.**   Compared to the best-reported accuracies of RGB-D Object [42] (99.51%) and ModelNet40 [43] (97.79%), MVP-N (91.61%) exhibits more room for improvement. Compared with ScanObjectNN [44] (90.74%) and FG3D [41] (93.99%, 83.94%, 79.47%), MVP-N has similar accuracy. However, the gap between the model performance and upper-bound accuracy for these two datasets is unknown owing to the lack of human evaluation. For MVP-N, since the human accuracy is 100% and SVAI is approximately 99%, the upper-bound accuracy is estimated to be 99%-100%, leaving a 7%-8% performance gap. Moreover, as the new methods improve performance, MVP-N can easily keep challenging by introducing more view combinations, additional views, and new objects.

## 6   Conclusion

This study proposes MVP-N, a new dataset for real-world multi-view object classification. The multi-view label noise in MVP-N provides a new perspective on the study of learning from noisy labels. Unlike general single-image classification tasks, this study investigates soft label methods in the multi-view case by making view-level predictions consistent with human judgments. Moreover, four multi-view-based feature aggregation methods and twelve soft label methods are benchmarked on MVP-N. Based on HPIQ annotations, this study analyzes the components in the feature aggregation modules and proposes a new evaluation protocol for soft label methods. Compared with multi-view-based feature aggregation methods, soft label methods incur a lower computational cost and can achieve better performance when the proportion of uninformative views is small. In future work, it would be better to propose new multi-view-based methods and estimate the amount of information utilized from each view to discriminate similar objects. Since many real-world applications embed a multi-camera system in edge devices without powerful GPU resources, the computation efficiency of multi-view-based methods should be considered. Such communication bandwidth and computation capability in a multi-camera system can be further used for evaluation.

**Broader impact.**   MVP-N can enable many real-world applications that require instance-level classification or similar-object differentiation, such as robotic grasping, auto-checkout in retail stores, and defect detection in manufacturing. Moreover, HPIQ annotations and the proposed MCDU metric can be applied to general single-image classification tasks. Specifically, single-image test sets can be divided into informative and uninformative sets based on HPIQ annotations. As analyzed in [80], label errors in test sets are pervasive and can destabilize performance evaluation. However, some label errors are caused by the intrinsic ambiguity of the image content (uninformative) and cannot be fully corrected. Therefore, MCDU would be a better option for evaluating model performance on uninformative sets than single-image accuracy.

**Ethical considerations.**   The hourly wages paid to the volunteer and annotators are above the minimum wage in South Korea. Collecting retail products from certain brands is permitted for research [81, 82, 83, 84].

## Acknowledgments and Disclosure of Funding

This work was supported by the National Research Foundation of Korea (NRF) grant funded by the Korean government (MSIT). (No. 2021R1G1A1094961 and 2020M3H6A1085498)

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
