# OpenReview forum: "MVP-N: A Dataset and Benchmark for Real-World Multi-View Object Classification"
_NeurIPS.cc/2022/Track/Datasets_and_Benchmarks — NeurIPS 2022 Datasets and Benchmarks _

### Official Review · Reviewer_VLRK · 2022-07-23
**265 Review**

**Rating:** 6
**Confidence:** 2

**Strengths:**

This work introduces data to solve a difficult object classification problem. This dataset likely has a sufficient number of samples to inspire diverse learning techniques, while still providing substantial validation and testing data. While the types of objects included in this dataset may be somewhat limited in form and function, this work inspires the application of fine-grained classification methods to any sort of similar object classification. Object classification at this granularity can be especially important to robotic grasping research.

The HPIQ annotations are an interesting concept that prompts further investigation. On top of the evaluation of informative/uninformative combinations, the HPIQ annotations inspire other research questions to investigate human thought processes to determine which areas of an object are most important for similar-object differentiation.

This work extensively evaluates other methods and datasets, including statistical analysis of performance with error margins.

Discussion of hyperparameters, compute hardware, supplemental work, availability of code and data contribute to this work's reproducibility and usability in the future. In-depth discussion of data collection procedures and setup facilitate expansions and reproducibility.

**Weaknesses:**

A main weakness of this paper is the clarity and quality of some technical descriptions. See "clarity" section of the review.

I do not understand the decision to make the validation and test sets as large as they are with respect to the training set. It seems that such a small training set is not sufficient for robust learning.

Conclusion, societal impacts, future work discussions are too brief.

Authors mention the noise in their dataset but do not elaborate on the benefits of having this noise.



**Additional Feedback:**

I will consider a stronger accept if you can more accurately explain the train/validation/test split decision and alleviate some of the technical description issues. Also, I think it is worth discussing the aforementioned dataset/annotation noise more and proposing uses for this noise in the "broader impacts" section.

It seems that this dataset has potential to enable solving of tough similar-item classification problems for everyday objects and may spur research into human inuition for object differentiation! I think that this dataset is a valuable contribution to the community, and despite the problems that I mentioned, I marginally accept.

**Clarity:**

The data collection methodology sections are very detailed, though I feel that my understanding of some other parts of the paper is limited due to inconsistencies in grammar and verbage. Namely, the parts of the paper that I feel should be reworked are: related work (soft label), training/validation/test split details and rationalization, novel soft label evaluation protocol, justification of conclusions, discussion of real-world deployment applications.

**Correctness:**

I feel that this dataset is constructed soundly, though the data collection methods (only a few object perspectives) and decisions to crop images to a relatively small image tile may limit future use cases of the dataset. The use of experienced human annotators and manual evaluation of annotation quality lends to the credibility of this dataset.

**Documentation:**

Code and data are available. Code implementation is modern and meant to run on modern hardware. Code implementation has some setup instructions. Paper & supplemental include details about data management/use, code setup, hyperparameters, etc. All of these things support the reproducibility of the work and expansion of the work in the future.

I did not find infomation about licensing or data maintenance. There can always be more documentation.

**Ethics:**

I have no ethical concerns with this work.

**Relation To Prior Work:**

It seems that this work is differentiated from previous work in a number of ways, including the composition of the dataset with respect to inter- and intra-class similarity, and the HPIQ annotations

**Summary And Contributions:**

The major contribution of this paper is a dataset that poses a challenging multi-view object classification problem. This dataset includes HPIQ annotations that reflect human intuition on what regions of an object may be important for differentiation of alike objects. In addition to this dataset, the authors also provide a benchmark of multiple existing feature aggregation methods and discuss their viability in real-world deployment scenarios. The authors evaluate the benchmark using a number of metrics, including a novel metric that they propose.

---

> ### Author Response · Authors · 2022-08-17
> **Response to Reviewer VLRK (Part 2)**
>
> **Q4: “Authors mention the noise in their dataset but do not elaborate on the benefits of having this noise. I think it is worth discussing the aforementioned dataset/annotation noise more and proposing uses for this noise in the "broader impacts" section.”**
>
> The multi-view label noise in the proposed dataset provides a new perspective on the study of learning from noisy labels. The noise can validate the robustness of soft label methods that make view-level predictions consistent with human judgments for multi-view object classification. The noise can also be generalized to the label noise caused by intrinsic ambiguity of image content (uninformative) in general single-image classification tasks. Since this label noise cannot be fully corrected, MCDU would be a better option to evaluate the model performance on uninformative images than single-image accuracy if HPIQ annotations are provided in test sets.
>
> We have added this part in the Conclusion and Broader Impact section in the revised paper.
>
> **Q5: “The data collection methods (only a few object perspectives) and decisions to crop images to a relatively small image tile may limit future use cases of the dataset.”**
>
> Two previous datasets [1, 2] collect multi-view images in the real capture environment. RGB-D Object [1] places the object on the turntable spinning at a constant speed and collects around 750 images with three cameras. MIRO [2] collects 160 images captured with 10 elevation angles and 16 azimuth angles. In this work, MVP-N defines various combinations of poses and rotations for the object and collects 1088 images with multiple cameras (12 or 16). Compared to RGB-D Object [1] and MIRO [2], MVP-N provides more object perspectives (views) in the data collection process.
>
> The resolution of captured images in MVP-N is 1920 × 1080. Since the commonly used resolution of views in previous datasets is 224 × 224, each image is cropped based on the center point of its bounding box annotation. This step removes most of the background and leaves the whole foreground object. RGB-D Object [1] also crops images, but the resolutions of the cropped ones are much smaller than 224 × 224 due to the low capture resolution (640 × 480). Moreover, as analyzed in [3], high-resolution views introduce more data communication cost and inference latency in distributed multi-camera systems, which limits real-world applications with a computational resource constraint, such as robotic grasping. Therefore, we think the resolution setting in MVP-N is rational and acceptable.
>
> Details of the data collection process for RGB-D Object [1] and MIRO [2] are provided in Appendix A.1.1 and A.1.3 (Supplementary Material), respectively. In addition, we have improved the clarity of the ‘Data collection’ and ‘Data preprocessing’ subsections in the revised paper.
>
> [1] A large-scale hierarchical multi-view rgb-d object dataset. (2011 ICRA)
> [2] Rotationnet for joint object categorization and unsupervised pose estimation from multi-view images. (2019 TPAMI)
> [3] Communication-efficient View-Pooling for Distributed Multi-View Neural Networks. (2020 DATE)
>
> **Q6: “I did not find information about licensing or data maintenance. There can always be more documentation.”**
>
> The license of the proposed dataset is Attribution-NonCommercial-NoDerivatives 4.0 International (CC BY-NC-ND 4.0).
> The DOI of the proposed dataset is https://doi.org/10.34740/KAGGLE/DS/2267231.
> The license of the code is MIT License.
> Data maintenance consists of two parts: error report and data update.
> 1. Error report: dataset download and usage issues, corrupted data files, and annotation errors.
> 2. Data update: new multi-view sets, real captured views, and objects will be introduced to keep MVP-N challenging.
>
> We have added this part in the revised dataset document (Supplementary Material).

---

> ### Author Response · Authors · 2022-08-17
> **Response to Reviewer VLRK (Part 1)**
>
> Authors thank the reviewer for the thoughtful review. We sincerely hope our response can address the concerns.
>
> **Q1: “A main weakness of this paper is the clarity and quality of some technical descriptions. The data collection methodology sections are very detailed, though I feel that my understanding of some other parts of the paper is limited due to inconsistencies in grammar and verbiage. Namely, the parts of the paper that I feel should be reworked are: related work (soft label), training/validation/test split details and rationalization, novel soft label evaluation protocol, justification of conclusions, discussion of real-world deployment applications.”**
>
> We have improved the clarity and quality of technical descriptions for the parts below in the revised paper.
> 1. Section 2 Related Work: Soft label.
> 2. Section 3 MVP-N: Dataset Design and Construction: The whole section.
> 3. Section 4 Benchmark on MVP-N: Evaluation for feature aggregation methods & Evaluation for soft label methods.
> 4. Section 6 Conclusion and Broader Impact
>
> **Q2: “I do not understand the decision to make the validation and test sets as large as they are with respect to the training set. It seems that such a small training set is not sufficient for robust learning. more accurately explain the train/validation/test split decision.”**
>
> Previous datasets are used for category-level object classification that identifies the category of a previously unseen object. The settings for previous datasets are 1) fixed (12 or 20) or arbitrary (4 to 12) number of views are sampled for each object, 2) an object has a single multi-view set containing all sampled views, and 3) each category has several objects, thus giving several multi-view sets. Unlike previous datasets, MVP-N is designed for instance-level object classification that identifies whether an object is the same object previously seen. Therefore, directly using the settings for previous datasets may cause insufficient learning and test bias due to only a single multi-view set per object.
>
> To ensure sufficient learning for real-world objects, we refer to the training set design from a recent work [1]. That work achieves around 90% category-level accuracy on RGB-D Object [2] by using 40 sampled views and five multi-view sets (five objects) on average per category. MVP-N has a similar design for each object compared to the above design for each category in [1]. We use 40 sampled views and ten multi-view sets for each object. As shown in Table 4, we can achieve around 90% accuracy, which means the training set in MVP-N ensures sufficient learning for real-world objects.
>
> Since limited multi-view sets may cause the test bias due to numerous view combinations for an object, we construct large-scale validation and test sets compared to the training set. This step aims to provide diverse view combinations with different proportions of informative views and reduce the test bias.
>
> In the revised paper, we have added the design rationalization in lines 163~167 (Section 3 MVP-N: Dataset Design and Construction: Train/valid/test split and design rationalization)
>
> [1] Learning canonical view representation for 3d shape recognition with arbitrary views. (2021 ICCV)
> [2] A large-scale hierarchical multi-view rgb-d object dataset. (2011 ICRA)
>
> **Q3: “Conclusion, societal impacts, future work discussions are too brief.”**
>
> We have enriched the content of the Conclusion and Broader Impact section in the revised paper.

---

### Official Review · Reviewer_bPoc · 2022-07-26
**The work is meaningful and well-designed. Besides an interesting dataset, it also provides benchmarks on the multi-view object classification.**

**Rating:** 7
**Confidence:** 3
**Clarity:** Yes, the paper is well written and ea…

**Strengths:**

1. The collected dataset is interesting and fair. Plus, a lot of details of the dataset are given.
2. A benchmark of the tasks is given on the collected dataset.
3. The paper is well written and well motivated.
4. The experiments are comprehensive with released code.

**Weaknesses:**

Some points are confusing and need to be clarified in the revision.
1. what is the relation between "16k real captured views, and 9k multi-view sets" in line 39？ Are they overlapped or are they independent of each other? It is better to clarify the relation between "16k" and "9k".

2. In line 46, what does "high inter-class view similarity causes uncertainty of class labels for uninformative views, which generates multi-view label noise" mean? Concretely, how the uncertainty exists in the uninformative views?

3. For the "bounding boxes" in line 131, since the dataset has 10 experts to annotate the bounding box, how does the final bounding box is decided among the annotated 10 bounding boxes for each object in a view.

4. What does "sufficiently informative, less informative, and uninformative" mean？ It is better to have definitions or even a figure to introduce the difference between these three concepts.

**Additional Feedback:**

Please refer to the weaknesses and ethics parts.

**Correctness:**

Yes, it is correct and the collection is in a sound way.


**Documentation:**

Yes.

**Ethics:**

Yes, I recommend the author discuss whether it is permitted to collect the objects in "certain brands."

**Relation To Prior Work:**

Yes, the authors provided a comprehensive discussion with an impressive table to compared with the prior works.

**Summary And Contributions:**

The work provides an interesting dataset for multi-view object classification. The dataset will benefit the related research. Besides the dataset, it also provides a benchmark for the multi-view object classification. The work also provides a new metric and evaluation protocol for the dataset.

---

> ### Author Response · Authors · 2022-08-17
> **Response to Reviewer bPoc**
>
> Authors thank the reviewer for the thoughtful review. We sincerely hope our response can address the concerns.
>
> **Q1: “what is the relation between "16k real captured views, and 9k multi-view sets" in line 39? Are they overlapped or are they independent of each other? It is better to clarify the relation between "16k" and "9k".”**
>
> The train/valid/test split design scheme in MVP-N (44 objects) consists of views sampling and multi-view sets construction. In views sampling, 40/160/160 (train/valid/test) views are sampled for each object. In multi-view sets construction, 10/100/100 (train/valid/test) multi-view sets are constructed for each object by combining views from the 40/160/160 (train/valid/test) sampled views, respectively. Therefore, the views in 9,240 (9k) multi-view sets are overlapped with 15,840 (16k) real captured (sampled) views.
>
> In the revised paper, we have improved the clarity of this part in lines 38 ~ 39 (Section 1 Introduction) and lines 154~156 (Section 3 MVP-N: Dataset Design and Construction: Train/valid/test split and design rationalization)
>
> **Q2: “In line 46, what does "high inter-class view similarity causes uncertainty of class labels for uninformative views, which generates multi-view label noise" mean? Concretely, how the uncertainty exists in the uninformative views?”**
>
> In MVP-N, different objects can appear similar or identical for specific views, referred to as high inter-class view similarity. In this case, humans cannot classify the object accurately from these views under the fine-grained (instance-level) object categorization. Here, the views with human uncertainty of class labels are denoted as uninformative views. The inconsistency between the one-hot manner of class labels and human judgment causes multi-view label noise.
>
> In the revised paper, we have improved the clarity of this part in lines 45~50 (Section 1 Introduction).
>
> **Q3: “For the "bounding boxes" in line 131, since the dataset has 10 experts to annotate the bounding box, how does the final bounding box is decided among the annotated 10 bounding boxes for each object in a view.”**
>
> Captured images are equally distributed to annotators. Each image is only annotated once by one annotator. Therefore, there is only one bounding box annotation for each object in a view. Two researchers check the quality of bounding box annotations, and annotators correct the missed and imprecise ones.
>
> In the revised paper, we have improved the clarity of this part in lines 129~130 (Section 3 MVP-N: Dataset Design and Construction: Data annotation).
>
> **Q4: “What does "sufficiently informative, less informative, and uninformative" mean? It is better to have definitions or even a figure to introduce the difference between these three concepts.”**
>
> The definitions of three options for information quantity judgment are provided as follows:
> (1) *sufficiently informative*: Distinctive appearance is included sufficiently in the view. The object can be classified correctly without additional information from other views.
> (2) *less informative*: Some part of distinctive appearance is included in the view. High classification accuracy cannot be guaranteed by using only this view.
> (3) *uninformative*: Distinctive appearance is not included in the view. Additional information from other views is required to classify the object correctly.
>
> In the revised paper, we have added the definitions of these three concepts in lines 133~138 (Section 3 MVP-N: Dataset Design and Construction: Data annotation) and a figure (Figure 2(b)) to introduce the difference.
>
> **Q5: “I recommend the author discuss whether it is permitted to collect the objects in "certain brands."”**
>
> We checked the Ethics Guidelines [1] and recently published dataset papers [2, 3, 4] that collect retail product objects. The proposed dataset follows ethical standards mentioned in [1], and collecting the objects in certain brands is permitted for research.
>
> We have added this ethical consideration in the revised dataset document (Supplementary Material) and will continue to pay attention to this potential ethics issue in the future.
>
> [1] https://neurips.cc/public/EthicsGuidelines
> [2] MVTec D2S: densely segmented supermarket dataset. (2018 ECCV)
> [3] Messytable: Instance association in multiple camera views. (2020 ECCV)
> [4] DexYCB: A benchmark for capturing hand grasping of objects. (2021 CVPR)

---

### Official Review · Reviewer_8zZx · 2022-07-28
**MVP-N: A Dataset and Benchmark for Real-World Multi-View Object Classification**

**Rating:** 6
**Confidence:** 4
**Correctness:** Yes, the claims are correct, however,…
**Clarity:** Yes.

**Strengths:**

1.	4 feature aggregation and 12 soft label methods are benchmarked on MVP-N.
2.	According to Table2, the proposed dataset includes validation set and view-level annotation.


**Weaknesses:**

1.	I cant find the definition of MCDU in this paper.
2.	There is no new feature aggregation method proposed in this paper.
3.	The multi-view object classification framework is very straightforward and there is no detail about how to train the network.
4.	There is no new proposed framework in this paper to exploit the advantage of this dataset (e.g. view-level annotation).


**Additional Feedback:**

Already mentioned in weakness.

**Documentation:**

Yes.

**Ethics:**

No.

**Relation To Prior Work:**

Yes, they show the difference in Table2.

**Summary And Contributions:**

In this paper, the authors propose MVP-N dataset for multi-view object classification to combine information from multiple views for multi-view object classification. Their contributions are proposing a real-world multi-view fine-grained dataset with HPIQ annotations, summarizing recent multi-view-based feature aggregation methods and a new metric.

---

> ### Author Response · Authors · 2022-08-17
> **Response to Reviewer 8zZx (Part 2)**
>
> **Q3: “The multi-view object classification framework is very straightforward and there is no detail about how to train the network.”**
>
> We summarize 36 multi-view feature aggregation methods and provide a unified multi-view object classification framework illustrated in Figure 1. Moreover, the unified framework also contains soft label methods for multi-view object classification (score fusion for view-level predictions). Although the framework is straightforward, it covers and represents existing methods well.
>
> Existing feature aggregation methods can be categorized into two-stage and three-stage strategies (hypergraph-based and part-based). Since the selection criteria of methods for the benchmark are representative, practical in real-world scenarios, and source code available, this paper benchmarks two-stage methods on the new proposed dataset.
>
> The training scheme of two-stage methods consists of single- and multi-view training. In the single-view training, the network is trained as general single-image classification tasks. In the multi-view training, variable-length views are first zero-padded to the max number of views and then batched together. Here, the network is trained with the feature aggregation module. Details about how to train the two-stage network are provided in lines 186~196 (Section 4 Benchmark on MVP-N: Training details). For soft label methods, the training details are the same as single-view training. Details of code implementation for network training are provided at https://github.com/SMNUResearch/MVP-N.
>
> In the revised paper, we have improved the clarity of training details for two-stage feature aggregation and soft label methods in lines 186~196 (Section 4 Benchmark on MVP-N: Training details).
>
> Since three-stage methods are not considered in the benchmark, we do not provide the training details of three-stage networks in this paper. Details are given as follows.
> (1)	Part-based: The region proposal network is first trained to detect generally semantic parts inside views under the processed segmentation benchmarks. Then, fix the parameters of the region proposal network and only update the parameters of the feature aggregation module in the multi-view training stage. The training loss consists of the multi-task loss in Faster-RCNN and cross-entropy loss. These two terms are combined with a balanced factor. Details of code implementation for network training are provided at https://github.com/liuxinhai/FG3D-Net.
> (2)	Hypergraph-based: Global features are first obtained from two-stage networks after the multi-view training. Then, a hypergraph is built based on the global features. Here, nodes represent global features, and edges are generated based on the Euclidean distance between nodes. Cross-entropy loss is used for training hypergraph neural networks. Details of code implementation for network training are provided at https://github.com/iMoonLab/HGNN.
>
> **Q4: “There is no new proposed framework in this paper to exploit the advantage of this dataset (e.g. view-level annotation).”**
>
> As the reviewer’s comment, a new proposed framework for feature aggregation was not proposed since this paper was submitted to the Datasets and Benchmarks Track.
>
> In this paper, we benchmark the existing framework (two-stage) on the new proposed dataset following the scope of this track. Unlike general single-image classification tasks, we propose a new framework to study soft-label methods for multi-view object classification based on the multi-view label noise and view-level annotations. The new proposed framework uses soft label methods to make view-level predictions closer to human judgments/perceptions and achieve high multi-view accuracy after score fusion.
>
> To exploit the advantage of this dataset, we also propose a new evaluation protocol based on view-level annotations and validate the robustness of 12 soft label methods for multi-view object classification.

---

> > ### Comment · Reviewer_8zZx · 2022-08-26
> > **I will change my rating to 6**
> >
> > The authors solved all of my concerns. The detail of the training process and evaluation metric are more clear in the revised version. Thanks for the effort.

---

> ### Author Response · Authors · 2022-08-17
> **Response to Reviewer 8zZx (Part 1)**
>
> Authors thank the reviewer for the thoughtful review. We sincerely hope our response can address the concerns.
>
> **Q1: “I cant find the definition of MCDU in this paper.” “there is a missing metric MCDU.”**
>
> Mean confidence difference between predictions and ground truths (MCDU) is given by: $MCDU = \frac{1}{N}{\sum_{i = 1}^{N}}(\max({p_{i}}) - p_{i}(y_{i}))$
> where $N$ is the number of wrong predictions for all uninformative views in the validation or test set, $p_{i}$ is the predicted probability distribution, and $y_{i}$ is the ground truth class label. MCDU is a new metric for performance evaluation on uninformative views. When MCDU is high, wrong predictions from uninformative views bring very negative impacts on the performance. Since predictions from uninformative views are not reliable, a lower MCDU indicates better performance.
>
> In the revised paper, we have improved the clarity of this part in lines 210~216 (Section 4 Benchmark on MVP-N: Evaluation for soft label methods).
>
> **Q2: “There is no new feature aggregation method proposed.”**
>
> As clarified in NeurIPS 2022 Datasets and Benchmarks Track [1], “algorithmic advances” are not required in this track. As mentioned in [2], “algorithmic advances” are recommended for the main conference submission. We carefully follow the scope of this track [3] and focus on “new datasets and benchmarks on new or existing datasets.” Therefore, proposing a new feature aggregation method is not covered in this dataset and benchmark paper.
>
> The main contributions of our paper can be summarized as follows: we first identify the limitations of existing datasets for multi-view object classification and summarize 36 multi-view-based methods concerning three properties of practicability. To resolve the limitations of existing datasets, we propose a new dataset with view-level annotations for developing practical multi-view-based methods. Since the new proposed dataset naturally provides multi-view label noise due to the inter-class view similarity, it also enables the study of learning from noisy labels in the multi-view case. We also benchmark 4 feature aggregation and 12 soft-label methods on the new proposed dataset. To evaluate the performance of soft label methods, we propose a new metric and evaluation protocol based on view-level annotations.
>
> We consider proposing new feature aggregation methods based on the new proposed dataset in future work.
>
> [1] https://neurips.cc/Conferences/2022/CallForDatasetsBenchmarks
> [2] “If the main contribution is a new dataset, benchmark, or other work that falls into the scope of the track (see above), then it is ideally reviewed accordingly. As discussed in our blog post, the reviewing procedures of the main conference are focused on algorithmic advances, analysis, and applications, while the reviewing in this track is equally stringent but designed to properly assess datasets and benchmarks. Other, more practical considerations are that this track allows single-blind reviewing (since anonymization is often impossible for hosted datasets) and intended audience, i.e., make your work more visible for people looking for datasets and benchmarks.”
> [3] “SCOPE. In addition to new datasets and benchmarks on new or existing datasets, we welcome submissions that detail advanced practices in data collection and curation that are of general interest even if the data itself cannot be shared. Data generators, reinforcement learning environments, or benchmarking tools are also in scope. Frameworks for responsible dataset development, audits of existing datasets, identifying significant problems with existing datasets and their use, or systematic analyses of existing systems on novel datasets that yield important new insight are also in scope.”

---

### Author Response · Authors · 2022-08-17
**Updates in the revised paper**

Authors thank all reviewers for their valuable and constructive comments. Authors appreciate that reviewers find the proposed dataset interesting and valuable. We have carefully studied the comments on weaknesses and revised our paper. We sincerely hope the revised paper and response can address the concerns. In the revised manuscript, the clarity and quality of technical descriptions have been improved, and major revisions are listed as follows:
1. Section 1 Introduction: The concept of multi-view label noise is explained.
2. Section 2 Related Work: The clarity of the ‘Soft label’ subsection is improved.
3. Section 3 MVP-N: Dataset Design and Construction: The clarity of the whole section is improved. Definitions of three options for information quantity judgment are added in the ‘Data annotation’ subsection. A figure (Figure 2(b)) is also added to show examples of human judgments for information quantity. The design rationalization of the train/valid/test split is explained more clearly in the ‘Train/valid/test split and design rationalization’ subsection.
4. Section 4 Benchmark on MVP-N: The clarity is improved in ‘Training details,’ ‘Evaluation for feature aggregation methods,’ and ‘Evaluation for soft label methods’ subsections. More explanations and an equation for the proposed MCDU metric are added.
5. The content of the Conclusion and Broader Impact section is enriched.
6. Dataset document: Ethical considerations for object collection and the information about data maintenance are added.

---

### Meta-Review · Area_Chair_FFKG · 2022-09-09

**Recommendation:** Accept
**Confidence:** 5

**Metareview:**

***Quality:***
* \+  The presented work has strong use-case for multi-view object classification with diverse  examples for 44 products which are  captured  in multiple views.
* \+  The proposed dataset is also benchmarked with 4 feature aggregation and  12 soft label methods on the dataset which promotes strong adoption.
* \-  The proposed dataset has relatively few categories compared to larger-scale datasets.

***Clarity:***
* \+ The final draft of the paper has added revisions that have addressed the reviewers concerns about clarity
* \- The paper still has lot of abbreviations that need to be connected.
* \+ The paper's description of the dataset is very detailed and clear

***Originality:***
* \+ Proposed dataset has been described by  reviewers as comprehensive in its collection of views and partitions
*  \- There are similar datasets released by Amazon and Alibaba that contain multi-instance product images which are larger scale.

***Significance:***
* \+ The reviewers' consensus is that the HPIQ annotations are interesting addition to the dataset to identify differentiating areas for an object.
* \+ The work is released on github with code and benchmarks which will ensure its reproducibility and reach.

---

### Decision · Program_Chairs · 2022-09-16

Accept